# Effectiveness and Safety of Transforaminal Spinal Endoscopy: Analysis of 1000 Clinical Cases

**DOI:** 10.3390/diagnostics15081021

**Published:** 2025-04-17

**Authors:** Ignazio Tornatore, Attilio Basile, Alessandro Aureli, Alessio Tarantino, Giuseppe Orlando, Rodrigo Buharaja

**Affiliations:** Ospedale Policlinico Casilino, 00169 Rome, Italy; attbasile@icloud.com (A.B.); aaureli.polcas@eurosanita.it (A.A.); alessio.tarantino1984@gmail.com (A.T.); gorlando10@gmail.com (G.O.)

**Keywords:** percutaneous transforaminal endoscopic surgery, discectomy, lumbar hernia, hernia L5 S1

## Abstract

**Background**: Transforaminal spinal endoscopy is a minimally invasive technique used to treat several spinal conditions. Thanks to its minimally invasive approach, it offers numerous advantages over traditional open surgery, including reduced blood loss, faster post-operative hospital discharge, quicker recovery, and lower complications. This study aims to evaluate the efficacy and safety of transforaminal spinal endoscopy on a large cohort of patients. **Methods**: We conducted a retrospective study of 1000 patients who underwent transforaminal spinal endoscopy between January 2015 and December 2023. All patients were treated by a single surgeon in a single hospital. The patients presented with persistent symptoms of radicular pain, neurological deficits related to herniated discs, and foraminal stenosis. All patients underwent transforaminal spinal endoscopy using a transforaminal approach. Clinical outcomes were evaluated using the VAS (Visual Analogue Scale) for pain, the ODI (Oswestry Disability Index) for functional disability, and patient satisfaction. Perioperative complications were recorded and analyzed. **Results**: Reduction in Pain**:** The mean VAS score decreased significantly from 8.2 ± 1.3 pre-operatively to 2.1 ± 1.5 at 12 months post-operatively (*p* < 0.001). Functional Improvement: The mean ODI improved from 56% ± 12% pre-operatively to 18% ± 9% at 12 months post-operatively (*p* < 0.001) Patient Satisfaction: 92% of patients reported a high level of satisfaction with their treatment results. Complications: Perioperative complications were minimal, with an overall common complication rate of 4%. No major complications or functional impairments were observed. **Conclusions**: Transforaminal spinal endoscopy is associated with good clinical outcomes and a low complication rate in patients with spinal pathologies. This study supports the adoption of this technique as a first-line treatment for selected patients, offering a less invasive and equally effective option compared to traditional surgery.

## 1. Introduction

Lumbar disc herniation (LDH) affects 1–5% of the global population annually with increasing incidence, making it a common and prevalent spinal disease [1]. In the U.S.A., 80% of people have reported at least one episode of lumbar spine disease in their lifetime [2]. The natural course of spinal disease is generally favourable, with most cases responding to conservative treatment. However, surgery may be indicated when conservative treatment fails or progressive neurological deficits develop [3]. Recent studies have reported that surgical treatment leads to a greater reduction in leg pain on long-term follow-up compared with conservative management for sciatica lasting from four to twelve months [4,5]. With the increasing concept of minimally invasive surgery and innovation in the design of surgical instruments, percutaneous endoscopic lumbar discectomy (PELD) has become the mainstream surgical technique for treating LDH due to its advantages, such as minimal trauma, shorter hospitalization, and rapid recovery [6,7]. PELD includes two different surgical approaches, percutaneous endoscopic transforaminal discectomy (PETD) and percutaneous endoscopic interlaminar discectomy (PEID), each with its own advantages and risks in treating LDH [8]. PETD removes the herniated disc through the “safety triangle”; as described by Kambin and Brager, the “safety zone is the area surrounding the superior endplate of the inferior vertebral body, superior articulating facet, and exiting nerve root (ENR)” (Figure 1) of the intervertebral foramina without laminectomy and dural retraction, causing less damage to the spinal canal and the soft tissues of the lumbar spine [9,10]. A recent study reported that PTED was non-inferior to open microdiscectomy in reducing leg pain. PTED resulted in more favourable outcomes for self-reported leg pain, back pain, functional status, quality of life, and recovery in many recent studies [11]. The diagnosis of lumbar disc prolapse or herniation is made by magnetic resonance imaging (MRI), which is the gold standard for diagnosing spinal disease due to its superior ability to discriminate between disc tissue and the spinal cord and reveal key information regarding other morphometric issues [12]. Initially, computer tomography scans provided the first view of the spine, but have since been replaced by MRI with 1.5 T. MRI studies in selected volunteers have shown up to a 36% incidence of disc protrusion and extrusion in patients without symptoms, thus highlighting their low predictive value for the development of back and leg pain [13]. Therefore, it is important to define the correlation between MRI findings and clinical features. This study aims to demonstrate the reduction in lumbar pain with the VAS and highlight the complications, with a particular focus on L5-S1. All interventions were performed by a single operator (I.T.) in the orthopedics and traumatology department of a single hospital (Policlinico Casilino, Rome, Italy). The contribution of each author was as follows: study design (I.T., A.A., and R.B.), performance of surgery (I.T., R.B, and G.O.), data collection (A.T. and A.B.), and data elaboration (A.T. and A.B.).

## 2. Material and Method

This is a retrospective study conducted at Policlinico Casilino in Rome (Italy) during the period between 1 January 2015 and 31 December 2023. In this retrospective study, we aim to present our experience with 1000 patients with lumbar pain surgically treated with PTED between 2015 and 2023, with all interventions performed by a single operator. We enrolled 1000 patients who were treated with transforaminal endoscopic discectomy. All patients were treated by the same surgeon and gave their informed consent to the processing of their personal data for medical–scientific purposes. We considered patients aged 30–60, a value necessarily broad due to the general population distribution of the pathology, while patients with neoplastic conditions or with co-existing severe spinal pathologies were excluded. We also excluded patients with mental pathologies in consideration of the need to guarantee constant and punctual follow-up, in addition to the possible risks linked to poor compliance on the part of the patient in the perioperative period. Each surgical intervention was planned in detail through the analysis of instrumental assessments (MRI and CT scan) in pre- and post-operative briefings scheduled on a weekly basis, while the hospitalization and discharge supervision were handled by the individual doctors on duty. The operating room staff used the safety devices required for exposure to radiation, which, although present in extremely small quantities, represented a factor to take into consideration given the large number of patients treated in a relatively short period of time. Patients were selected according to the inclusion and exclusion criteria, as stated in Table 1.

All 1000 patients enrolled were treated with percutaneous transforaminal endoscopic discectomy. Prior to the operation, all patients were evaluated using magnetic resonance imaging (MRI), and dynamic X-ray images were used to examine spinal instability or the backward slippage of the vertebral body. For the examination, we requested MRI with a 1.5 T scanner. All patients underwent an assessment using the VAS (Visual Analogue Scale) pre-operatively and 1 month, 3 months, 6 months and 12 months after surgery. Furthermore, patients underwent an assessment using the ODI pre-operatively and 1 month, 3 months, 6 months and 12 months after surgery. Furthermore, all patients were subjected to a satisfaction questionnaire at 12 months, which ranged from 1 to 10, with 1 being very bad and 10 being maximum satisfaction. In addition to assessments at pre-established endpoints, all patients were given the possibility of being able to quickly contact a member of staff in case of urgent needs or simply for any questions. To date, none of the patients have taken advantage of this possibility.

### 2.1. Surgical Method (L1/L5) for Transforaminal Endoscopic Discectomy

General intravenous anesthesia (GIVA) was routinely used. The exceptions were patients whose general condition contraindicated GIVA; they were sedated with local spinal anesthesia for lumbar fixation. Patients were positioned prone on a radiolucent bed, with special supports at the level of the iliac crests and chest (Figure 2). A C-arm fluoroscopy machine was used. The arms were abducted and placed on padded boards, and the legs were kept in extension. A standard dose of Cefazolin 2 g was administered intravenously before the skin incision. The role of C-arm fluoroscopy is crucial at this phase. The surgical procedure can be started only after obtaining a perfect anteroposterior and lateral image of the vertebral level. This particular step of the procedure required a standardization procedure through instructions on the positioning and mobilization of the device, as the radiology staff in charge of the operating room had no experience regarding endoscopic spinal surgery procedures. The needle entry point was selected 8–12 cm from the midline, situated just above the facet joint on the lateral view. A 22-gauge needle was inserted into the disc, followed by the injection of contrast medium (7 mL of iohexol with 3 mL of methylene blue) into the disc. The correct setting of the entry points represents a key step in the entire procedure. A blunt tapered cannulated obturator was passed over the guide wire under fluoroscopic guidance. Sequential protective cannulas were introduced over the obturator until the final protective cannula was in position. Foraminoplasty was performed using a 10 mm diameter trephine via the transforaminal approach. The protective cannula was then replaced with an 8 mm working cannula. The endoscope (Endofest Luma) was positioned through a working casing pipe inserted through a 1 cm skin incision centred on a guidewire (Figure 2). The prominent intervertebral disc, stained with methylene blue, was removed using a pituitary rongeur under the endoscope (Figure 3). Next, the hypertrophied ligamentum flavum, facet joints, and anterior herniated disc were resected to achieve decompression. Epidural bleeding was controlled with a radiofrequency probe (Endofest Luma srl, Palermo, Italy) under physiological solution irrigation. Finally, the working cannula and endoscope were removed, and the skin was sutured.

### 2.2. The Positioning of Patient for L5-S1 Endoscopic Discectomy

L5-S1 herniations represent approximately 40–50% of all herniated discs, although this value depends on the studies considered. Percutaneous endoscopic discectomy often presents a difficult challenge for the surgeon, especially those with less experience, due to anatomical considerations. The iliac wing frequently impedes optimal entry, risking the outcome of the operation, particularly in cases of downward-migrated hernias. We believe that correct patient positioning on the operating bed can facilitate this procedure, representing an essential parameter in the pre- and intra-operative management of the patient and an element whose standardization represents a necessity for the success of the surgical procedure. In all cases considered for this study, patients with an L5-S1 hernia were positioned prone. Rolls of soft sheets approximately 5 cm thick were positioned under the chest and at the level of the iliac crests. Additional padding was placed at the level of the legs and feet to maintain them at approximately 30–40°. The iliac crest was positioned at the level of the folding point of the bed, and the lower limbs were bent by approximately 40–45°. Furthermore, the patient was positioned at a 20° Trendelenburg tilt. Subsequently, the patient was positioned with a lateral bend—lower limbs to the left if the endoscope entrance was on the right, and lower limbs to the right for entrances on the left. This position allows for a modification of the iliac wing height and the greater opening of the foramen, facilitating easier entry and contributing to better surgical outcomes. The choice of this positioning and the angle at which the entry takes place allows for a greater ease of handling in the surgical movement and guarantees the best safety profile in relation to the possibility of damaging the local vascular–nervous structures. The photos below describe this position. The techniques just discussed are well exemplified in the photos, which were acquired during some of the procedures of the patients in the study, subject to the granting of due consent and relative authorization for publication. The surgical technique proceeded as previously described. The operating room nursing staff were not selected on the basis of specific skills; we simply collaborated with staff with general training in OR routine. Even the radiological technicians present during the interventions did not have specific training but were trained progressively through pre- and post-operative briefings. The anesthesiologists in the room had no experience in assisting with this particular procedure before the start of the study, nor were they given specialized training.

## 3. Results

Of the 1000 patients enrolled in this study, the average pre-operative VAS score was 7.7 at the lower limb level and 6.7 at the lumbar level (Figure 4). This value was drastically reduced to 2.7 for the lower limbs and 3.2 for the lumbar spine after one month. At six months, the values were 2.4 for the lower limbs and 2.8 for the lumbar spine. At twelve months, the values further decreased to 1.9 for the lower limbs and 3.0 for the lumbar spine. No significant differences were recorded between the two sexes or the various endoscopies used. There were no deaths among patients, nor were there any patients lost to follow-up. No statistically significant differences were discerned based on the anthropometric or clinical parameters examined in the patient recruitment operations, nor were any complications unrelated to the surgical procedure resulting from previous medical conditions reported.

Of the 1000 patients enrolled in this study, their pre-operative ODI was 0.78. After one month, this value decreased to 0.44, at six months to 0.3, and after one year to 0.16 (Figure 5). In this case, there were no differences between the two sexes or based on the different endoscopies used. Again, no statistically significant differences were discerned based on the anthropometric or clinical parameters examined. Among the 434 patients surgically treated for the L5-S1 segment, the pre-operative VAS score was 8.1 at the lower limb level and 7.3 at the lumbar level. After one month, the VAS score dropped drastically to 3.4 for the lower limbs and 3.7 for the lumbar region. At six months, the scores were 2.2 and 2.5, respectively, and at twelve months, they were 1.8 for the lower limbs and 2.8 for the lumbar region.

The satisfaction questionnaire was completed by all 1000 patients, with an average satisfaction score of 9.3 after one year. For patients treated at the L5-S1 level, the satisfaction score remained at 8.7, values that we consider extremely high and confirmed in all subsequent endpoints considered. The trends of all the endpoints considered were towards a reduction in pain, without any recrudescence or acute episodes being observed at a distance. During our study, we also encountered complications: 58 patients required a repeat operation at the same level within 30 days due to the incomplete removal of the hernia. Many of these cases occurred during the initial phase of endoscopic procedures. As the number of cases performed increased, the rate of re-operations decreased, and in the last 200 cases performed, there were no re-operations. Additionally, we observed five cases of infection and no cases of dural tear or nerve injury. In all cases included in the study, no complications or adverse events related to bad practice were appreciated by the healthcare support staff (nurses and radiologic technicians).

## 4. Discussion

The results of this study demonstrate that percutaneous transforaminal endoscopic discectomy can be an effective treatment for patients with lumbar disc herniation. Significant improvements in ODI and VAS scores were observed, with very few complications [14]. The rapid development of high-resolution endoscopes has increased interest in minimally invasive spine surgery technology and significantly expanded the use and proposed indications for this procedure. With more emphasis now on enhanced recovery after surgery, PELD has shown favourable long-term outcomes in the results obtained and presented in the literature by all the authors considered in this study [15]. The integrity of the paraspinal muscle was preserved by directly reaching the target position using a muscle-splitting technique with sequential dilators and a blunt obturator [16]. A recently published meta-analysis compared PTED with open microdiscectomy in the treatment of sciatica [17]. This meta-analysis included 14 prospective studies, of which 9 were randomized, and eventually concluded that moderate-quality evidence existed for no difference in leg pain reduction or functional status at long-term follow-up with extremely solid scientific validity thanks to the scientific rigour of the investigation methodology used. Gibson and colleagues also compared both techniques in a study with a sample size of 143 patients, showing the superiority of the transforaminal endoscopic discectomy technique in improving the VAS of leg pain at the end of a 2-year follow-up, obtaining extremely encouraging data and giving new impetus to the extension of the use of the method and the study of its possible further applications, although the study methodology used, the number of patients examined and the overall follow-up period could have at least partially compromised the validity of the data collected in terms of scientific evidence [18]. PTED, as reported in the literature, is associated with better outcomes, small incisions, less damage to human tissues, lower complication rates, and shorter hospitalization times [19,20]. In our centre, approximately 90% of lumbar discectomy operations are currently performed endoscopically, with only about 10% performed via open microdiscectomy. This percentage was completely reversed about 10 years ago with a gradual but progressive reversal of the trend observed over the proposed period with a growing increase in the number of cases considered. The complications described appeared in the first cases, showing that with a good learning curve, the complications are significantly reduced. The results obtained in radiologic he pain assessment also represent a significant advantage for the proposed method, which is already intrinsically less invasive compared to traditional surgery. The data obtained in fact highlight a clear improvement in painful symptoms and an extremely gratifying picture in relation to patient satisfaction, which is appreciable in all pre-established endpoints and which is maintained throughout the entire period of the time considered. Furthermore, we believe that the progressive standardization and optimization of pre- and post-surgical procedures (acceptance, hospitalization, post-operative management, and discharge) has contributed to minimizing possible complications and patient discomfort, allowing the precise planning of hospitalization times. None of the patients treated required extra days of hospitalization compared to what was established and proposed to them in the informed consent. The limitations of our study are that all operations were performed by a single operator and in the same centre, reducing the reliability of the data obtained in relation to the standardization of procedures. We believe that the rate of acute complications (within 30 days) is attributable to the failure to perfect the various steps of the procedure, but, once standardized, complications were no longer reported and there were no indications for new surgical interventions. In conclusion, percutaneous endoscopic lumbar discectomy (PELD) as a minimally invasive procedure, offers several advantages over conventional open surgery, including reduced operation time, decreased blood loss, minimal soft tissue damage, and expedited return to work. Nonetheless, it also presents several challenges. Primarily, it necessitates a high level of expertise in spinal surgery. The incidence of complications or the need for revisions among novice surgeons may be elevated due to a lack of experience and proficiency, contributing to a steep learning curve [21]. Also, the percutaneous transforaminal endoscopic discectomy (PTED) approach appears to yield favourable outcomes for the treatment of lumbar spinal stenosis (LSS). Nevertheless, this technique may be less efficacious for LSS patients who exhibit lumbar instability or necessitate revision surgery in the same segment [22].

## 5. Conclusions

Transforaminal endoscopic discectomy in our experience showed good clinical outcomes regarding the VAS for back pain, the VAS for leg pain, and the ODI score with few complications. In our hospital, endoscopic treatment for lumbar hernias currently represents the treatment of choice, surpassing open microdiscectomy, with undeniable benefits for patients in terms of pain, functional recovery and quality of life after surgery. Further studies with higher levels of evidence (RCT and meta-analysis) should be conducted to ensure the greater validity of the evidence obtained, possibly involving other structures and with larger operational cases. We are aware that the technique we use in our practice requires a significant learning curve; however, we believe that, under adequate supervision, it is possible to master it with a reasonable number of cases, minimizing the possible risks linked to recurrences and intra- and post-operative complications. We also believe that the lack of complications relating to the management of the other professional figures present in the room (nurses and radiologic technicians) denotes a positive impact on the amount of resources necessary to adequately train staff in the management of the operating room during this particular surgical procedure. We are aware that our study has objective limitations, in particular regarding the case studies involving a single operator and at a single centre; however, we believe that the super-specialist nature of the proposed treatment, combined with the large number of cases examined, contributes to giving scientific credence to the data collected. Although a final endpoint at 2 years may be limiting, the post-operative evaluation continues today with periodic evaluations, and it is our intention to continue observing these patients to acquire data in the long term, integrating the data obtained in future studies.

## Figures and Tables

**Figure 1 diagnostics-15-01021-f001:**
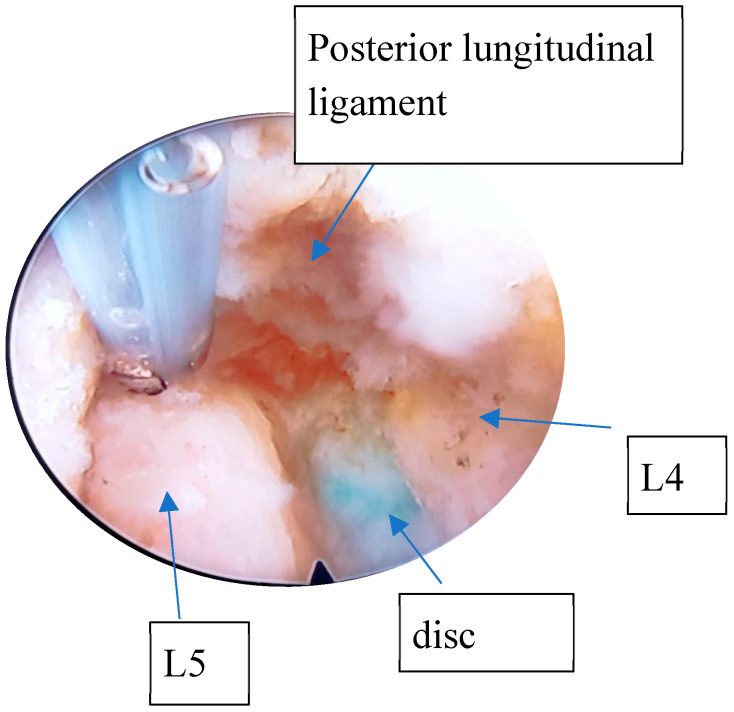
Endoscopic view during discectomy of L4–L5.

**Figure 2 diagnostics-15-01021-f002:**
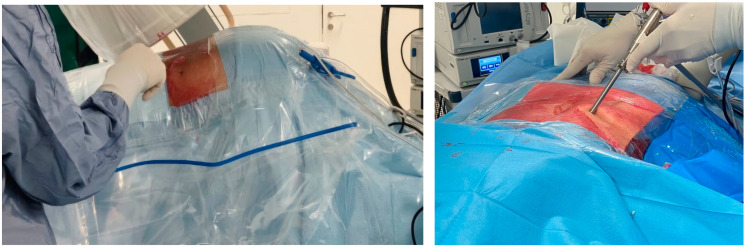
Representation of position of patient for L5-S1 PTED.

**Figure 3 diagnostics-15-01021-f003:**
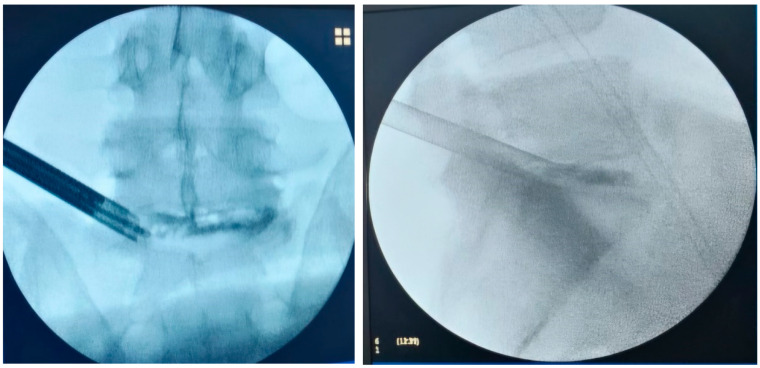
Intra-operative fluoroscopy in AP and LL of L5-S1 PTED.

**Figure 4 diagnostics-15-01021-f004:**
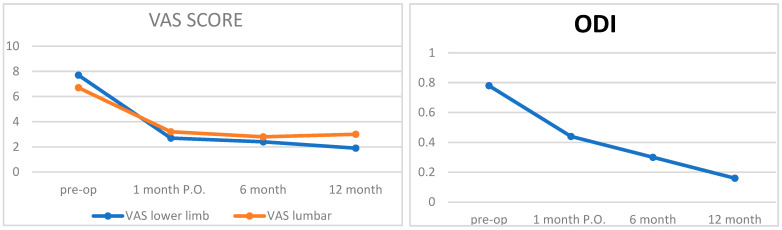
Vas score progression; ODI score progression.

**Figure 5 diagnostics-15-01021-f005:**
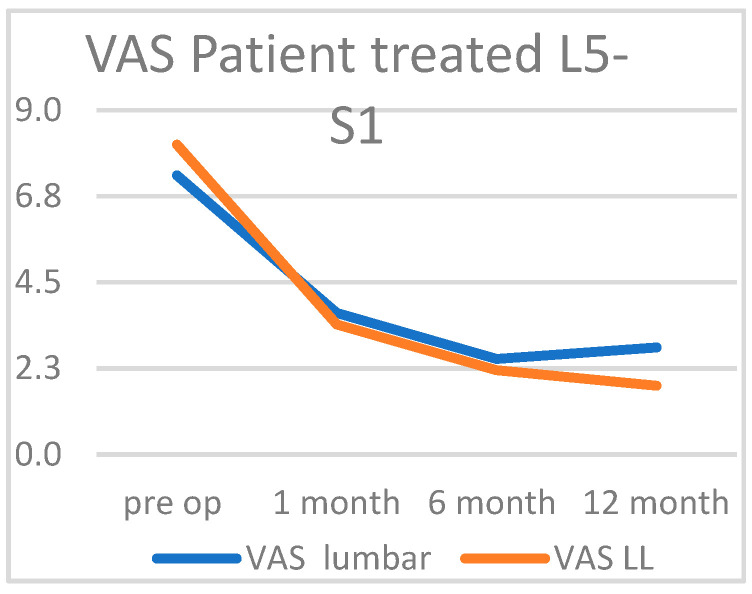
Vas in patient treated L5-S1.

**Table 1 diagnostics-15-01021-t001:** Inclusion/exclusion criteria.

Inclusion Criteria	Exclusion Criteria
neurogenic intermittent claudication	potential mental illness
radicular irritation with or without sensory loss	multilevel lumbar spinal stenosis
concordant imaging diagnosis of lumbar stenosis or spinal disc herniation	severe scoliosis
failure of conservative treatment for at least 3 months	patient with spinal/vertebral tumour or vertebral tumor
age between 30 and 60	active neoplastic systemic pathologies

## Data Availability

The data are available on request from the authors.

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
