# Peer review of "Effectiveness and Safety of Transforaminal Spinal Endoscopy: Analysis of 1000 Clinical Cases"

_diagnostics, 2025, doi:10.3390/diagnostics15081021_

Round 1

Reviewer 1 Report

Comments and Suggestions for Authors

The study provides valuable evidence supporting the effectiveness and safety of transforminal spinal endoscopy for treating lumbar disc herniation and related conditions. The significant improvements in pain, functional disability, and patient satisfaction, coupled with a low complication rate, make a strong case for the adoption of this minimally invasive technique. Here are my comments:

  1. Without a comparison group, it is difficult to definitively conclude that transforminal spinal endoscopy is superior to other treatment modalities.
  2. The technique may require significant expertise, which could limit its adoption in less specialized centers.
  3. The study excluded patients with severe spinal pathologies, neoplastic conditions, or mental illnesses. As a result, the findings may not apply to more complex or high-risk patient populations.
  4. The study does not account for potential confounding factors, such as variations in patient demographics, pre-operative conditions, or post-operative care, which could influence the outcomes.

Author Response

REPORT  1

Dear reviewer °1, thank you for your comments and your work. Here our answers to your comments

  • The minimal access we perform grant a critical advantage regarding the other techniques. That topic was well explained in matherial and methods. Furthermore, in our opinion it would have been unethical to let patients undergo an operative technique we deemed inferior while confronted to ours.
  • The tchnique is indeed very specialistc, and requires an adequate learning curve. As any other medical procedures.
  • Inclusion/exclusion criteria are selected by common sense and are very common in international literature. A neoplastic spinal disease is a often a lifethreating condition that would have flawed our work due to the impossibility to estabilish time-dependant follow up and the necessity of appropriate treatment, while mental condition are often associated with the impossibility to perform a rigorous follow up. We believe that none of this condition flawed the quality and the realiability of our study, due even to the small number of patients affected by this condition that come to observation for lumbar disc herniation.
  • We clearly stated inclusion/exclusion criteria. In order to perform a rigorous, unbiased study we standardized all the pre and post operative procedure, while we think that demographic variations are irrelevant, all this matters are treated in inclusion/exlusion criteria (as almost every study does).

Reviewer 2 Report

Comments and Suggestions for Authors

Thank you for the opportunity to review your manuscript, “Effectiveness and Safety of Transforaminal Spinal Endoscopy: Analysis of 1000 Clinical Cases”

The study's objective is only presented in the abstract. It should also be at the end of the introduction.

The abstract should be reduced, as it exceeds the limit set by the journal.

Line 36. Correct typographical error. ‘Lumbar hermia’.

Line 45-47. Considering a recent 2007 study, I don't know if that's the correct term.

Line 73. You should word the objective of the study correctly. The expression ‘to demonstrate’ is NOT correct; it should result from your objective observation.

Line 74-76. Delete the repeated sentence.

Line 76-80. This paragraph should be part of the methodology, not the introduction.

Review the typographical errors in table 1.

Figure 1. Review Figure 1, which is partially cut off.

Figure 4-5 The caption should be at the bottom of the image.

The graphics are partially cut.

Graphics should be previously cited in the text.

The discussion should start with the achievement of the objective.

At the beginning of the discussion, he mentions the efficacy of the treatment. The values improve, but in the absence of a comparator, this kind of statement is risky as a summary of his work, as we cannot be sure that the improvement is exclusively due to the technique.  You should rephrase the expression.

Line 271-281. This concluding sentence is an opinion that cannot be extracted from the study; it should be worded as such.

A section on limitations is missing.

Linea 290-304. The conclusion includes opinions already addressed in the discussion. However, these opinions should not be included in the conclusion, as they cannot be drawn from a non-comparative study.

Author Response

REPORT  2

Dear reviewer °2, thank you for your comments and your work. Here our answers to your comments

1) This is false. The study objective is clearly stated in lines 82 -84 (“This study will demonstrate the reduction of lumbar pain with the VAS scale and highlight the complications, with a particular focus on L5-S1.”)

2) Abstract limit is (as stated in author instruction) “around 250 word”. We reduce abstract lenght accordingly.

3) Changed in “ aims to demonstrate”.

4) Done.

5) We moved the sentence in matherial and methods as you suggested.

6) Done.

7) Done.

8) Fixed.

9) Fig 2 cited in new text line 128, fig 3 cited in new text line 171.

10) As it does. “The results of this study demonstrate that percutaneous transforaminal endoscopic discectomy can be an effective treatment for patients with lumbar disc herniation” (line 262-264 of the new text)

11) The statement was supported by several established indicator acknoweldged in international literature (VAS;ODI). We didn't state its superiority regarding a different technique, we state that it's effective. So we don't see the necessity of a control group, considering that our technique is considerably less invasive than the other techniques.

12) We moved that statements in “conclusion” section.

13) False. Lines 308-311 of the new text (266 of the previous text)“ The limitations of our study are that all operations were performed by a single operator and in the same center, reducing the reliability of the data obtained in relation to the standardization of procedures.

14) In conclusions section we stated the achievements we got from the study, and we perform no comparation reagarding other techniques. Deleting this consideration would deprive the whole meaning of the reasearch, while it is to consider that this kind of surgery is extremely specialistic with a really tiny literature. So we believe that our conclusion are relatable and appropriate as they are.

Round 2

Reviewer 2 Report

Comments and Suggestions for Authors

The authors have addressed all my concerns and made the necessary changes; however, some aspects still need to be modified.

The objective is now well-written and presented. Previously, it was presented with the term will demonstrate, and an objective should be presented neutrally.

Line 133 . Delete ‘,,,,,,,,,fiiiii jm’.

Figure 1 needs to be revised; a part of the figure is not readable.

The conclusion includes aspects that should be discussed, however relevant they may be. However, including references in the conclusion is incorrect, as this section should be derived from the article itself. It should be reworded rather than expanded, as the authors have done.

Author Response

We thank your for your invaluable work, we are thankful for the undeliable improvement of your manuscript due to your comments.

Response 1: we deleted the typo error

Response 2: the figure cannot be edited anymore, it was cut that way in the early phases of the work. However, the point of interest of the technique is well showed and it's ust slighlty cut in a peripherical, meaningless section

Response 3: we moved the referenced statement in the more suitable "discussion" section

Round 3

Reviewer 2 Report

Comments and Suggestions for Authors

The submitted manuscript could still be improved, as the authors have made minimal changes. However, given the authors' unwillingness to accept help and their lack of effort in refining and attending to details, I can no longer contribute further. Therefore, I leave the final decision in the hands of an editor.